# A Novel Systematic Oxidative Stress Score Predicts the Survival of Patients with Early-Stage Lung Adenocarcinoma

**DOI:** 10.3390/cancers15061718

**Published:** 2023-03-11

**Authors:** Jia-Yi Qian, Yun Hao, Hai-Hong Yu, Lei-Lei Wu, Zhi-Yuan Liu, Qiao Peng, Zhi-Xin Li, Kun Li, Yu’e Liu, Rang-Rang Wang, Dong Xie

**Affiliations:** 1Department of Thoracic Surgery, Shanghai Pulmonary Hospital, School of Medicine, Tongji University, Shanghai 200433, China; 2033265@tongji.edu.cn (J.-Y.Q.); wull_7@yeah.net (L.-L.W.); saklizhixin@163.com (Z.-X.L.); leeq8110@163.com (K.L.); 2School of Medicine, Tongji University, Shanghai 200092, China; 1810442@tongji.edu.cn (Y.H.); 1911469@tongji.edu.cn (H.-H.Y.); 2131068@tongji.edu.cn (Z.-Y.L.); 2011701@tongji.edu.cn (Q.P.); yueliu@tongji.edu.cn (Y.L.); 3Huadong Hospital Affiliated to Fudan University, Shanghai 200040, China

**Keywords:** stage I lung adenocarcinoma, epidermal growth factor receptor, prognosis, reactive oxygen species, oxidative stress

## Abstract

**Simple Summary:**

A retrospective study was performed on 955 eligible patients with stage I lung adenocarcinoma (LUAD) after surgery. The systematic oxidative stress score (SOS) was established based on three biochemical indicators, including serum creatinine (CRE), lactate dehydrogenase (LDH), and uric acid (UA). SOS is an independent prognostic indicator for stage I LUAD. In addition, the constructed nomogram based on SOS could accurately predict the survival of those patients.

**Abstract:**

This study aimed to construct an effective nomogram based on the clinical and oxidative stress-related characteristics to predict the prognosis of stage I lung adenocarcinoma (LUAD). A retrospective study was performed on 955 eligible patients with stage I LUAD after surgery at our hospital. The relationship between systematic-oxidative-stress biomarkers and the prognosis was analyzed. The systematic oxidative stress score (SOS) was established based on three biochemical indicators, including serum creatinine (CRE), lactate dehydrogenase (LDH), and uric acid (UA). SOS was an independent prognostic factor for stage I LUADs, and the nomogram based on SOS and clinical characteristics could accurately predict the prognosis of these patients. The nomogram had a high concordance index (C-index) (0.684, 95% CI, 0.656–0.712), and the calibration curves for recurrence-free survival (RFS) probabilities showed a strong agreement between the nomogram prediction and actual observation. Additionally, the patients were divided into two groups according to the cut-off value of risk points based on the nomogram, and a significant difference in RFS was observed between the high-risk and low-risk groups (*p* < 0.0001). SOS is an independent prognostic indicator for stage I LUAD. These things considered, the constructed nomogram based on SOS could accurately predict the survival of those patients.

## 1. Introduction

Lung cancer is one of the most common malignancies in the world and the leading cause of cancer-related death worldwide [1,2]. In recent years, more cases of early-stage non-small cell lung cancer (NSCLC) can be detected due to the popularity of chest computed tomography (CT) [3,4]. However, even after radical surgery, the prognosis of these patients remains heterogeneous [5,6,7]. In addition to the factors known to affect the prognosis of patients, such as subtypes of tumor and visceral pleural invasion (VPI) [8,9,10], some subtle factors in the human body may also play a role.

Emerging studies have suggested that most cancers can be attributed to long-term chronic inflammatory diseases [11,12]. However, the mechanisms by which inflammation drives tumor formation, growth, and metastasis remain unknown. Recent studies have shown that reactive oxygen species (ROS) is strongly associated with inflammatory responses [13]. Simply put, inflammatory cells release large amounts of ROS and secrete cytokines which also promote the production of ROS by adjacent cells. In addition, ROS can regulate multiple transcription genes, further enhancing the expression of proinflammatory cytokines [14].

ROS generation exceeding the ROS scavenging ability was defined as oxidative stress. Excess ROS can lead to oxidative damage to major components of living cells, including proteins, lipids, and DNA, ultimately leading to a wide range of pathophysiologies such as sepsis, aging, obesity, and cancer [15,16]. At present, several studies have shown that ROS may enhance genomic instability and promote malignant cell proliferation, tumor angiogenesis, invasion, and metastasis [17,18], but the relationship between systemic oxidative stress and the prognosis of patients with early-stage NSCLC is unclear. To date, there is no effective method for detecting ROS levels in clinical applications. Therefore, we included several relevant indicators that have been proposed by previous studies to reflect oxidative stress in patients, including total bilirubin (TBIL), direct bilirubin (DBIL), lactate dehydrogenase (LDH), creatinine (CRE), uric acid (UA), and serum albumin (ALB) [19,20,21].

This study aims to explore the relationship between biomarkers associated with oxidative stress and the prognosis of patients with stage I lung adenocarcinoma (LUAD). Moreover, a clinical model was constructed based on the results of this study to predict the long-term outcomes of these patients, which would help clinicians identify the high-risk population and promote more individualized treatment and follow-up.

## 2. Materials and Methods

### 2.1. Data Sources and Study Population

The patient data came from Shanghai Pulmonary Hospital and was approved by the Ethics Committee of Shanghai Pulmonary Hospital. In this study, we recruited patients with primary NSCLC who underwent surgical treatment at Shanghai Pulmonary Hospital between 2015 and 2016. All patients were evaluated for receiving any diet or antioxidant therapy before the surgery. The inclusion criteria were: (1) confirmed as LUAD; (2) no metastasis to the lymph nodes or other organs; (3) presence of one primary tumor only; (4) tumor size was between 0 and 4 cm; and (5) received radical resection. The exclusion criteria were: (1) pathology of adenocarcinoma in situ (AIS) or minimally invasive adenocarcinoma (MIA); (2) invasion of parietal pleura, vessels, or ribs; (3) age < 18 or age > 80 years; (4) received neoadjuvant therapy; and (5) perioperative death (died within 1 month after surgery).

### 2.2. Data Collection and Treatment

We collected information on the patient’s gender, age at the time of surgery, smoking history, surgical procedure, predominant pattern, tumor size, VPI, lymphovascular invasion (LVI), spread through air space (STAS), epidermal growth factor receptor (EGFR) mutation status, and adjuvant chemotherapy (ACT). We also obtained oxidative stress-related biochemical indicators such as TBIL, DBIL, ALB, UA, CRE, and LDH from the patients 3 days before surgery and used X-tile software to determine the optimal cut-off values. The optimal cut-off values for biochemical indicators were as follows: TBIL: 7.9 µmol/L, DBIL: 2.9 µmol/L, ALB: 38 g/L, UA: 325 µmol/L, CRE: 58 µmol/L, and LDH: 198 U/L. Biochemical indicators above the optimal cut-off value were defined as high-level, and vice versa as low-level. The pathological stage was determined according to the AJCC/UICC 8th edition.

### 2.3. Variable Declaration

According to the classification introduced by the International Association for the Study of Lung Cancer (IASLC), American Thoracic Society (ATS), and European Respiratory Society (ERS), histologic patterns of adenocarcinoma could be majorly classified as lepidic, acinar, papillary, micropapillary, and solid. The predominant histologic pattern was the pattern with the highest percentage and tumors. Tumors were then collapsed into three groups: lepidic predominant, acinar/papillary predominant, and micropapillary/solid predominant. In our study, AIS and MIA were excluded. EGFR mutation was detected by liquid/tissue biopsy, mainly including deletions in exon 19 (19-del) and a recurrent point mutation in exon 21 (L858R).

### 2.4. Follow-Up and Outcome

In this study, all patients were followed up after the resection of the primary tumor. The relevant information about postoperative patients was obtained through telephone calls or medical records. The follow-up duration ranged from 3.0–91.7 months, with an average of 56.2 months. Recurrence-free survival (RFS) was defined as the time from the date of surgery to the date of the first recurrence or last observation. Recurrence was confirmed by tissue biopsy or detailed examinations, including brain magnetic Resonance Imaging (MRI), chest CT, bone scan, or positron emission tomography-computed tomography (PET-CT).

### 2.5. Statistical Analyses

Categorical variables were tested by using Pearson chi-squared test and Fisher’s exact test. The *t*-test and Mann-Whitney U test were used to compare continuous variables in all cohorts. Univariable and multivariable Cox regression risk proportion models were used to identify independent prognostic predictors affecting patient survival and to calculate the corresponding Hazard ratio (HR) and 95% confidence intervals (CI). In order to extend the analysis, variables with *p* < 0.2 in the univariable Cox regression analysis were eventually included in the multivariable Cox regression analysis to obtain independent prognostic predictors associated with the RFS of patients. Statistical significance was considered as a *p* < 0.05 on two sides.

The cut-off values for RFS of six oxidative stress-related biochemical indicators, including TBIL, DBIL, ALB, UA, CRE, and LDH, were calculated by X-tile software (Copyright: Camp/Rimm, Yale University). According to the calculated cut-off values, all indicators were divided into high-level and low-level groups. Based on the results of multivariable Cox regression analysis, three biochemical indicators, including UA, CRE, and LDH, were identified, and the corresponding regression coefficients were finally determined. The formula of the systematic oxidative stress score (SOS) was as follows: SOS = sum (corresponding regression coefficient × status of biochemical indicator). The optimal cut-off value of SOS was analyzed by X-tile software, and the patients were also classified into low-level and high-level groups.

Nomogram was constructed by R version 4.1.1 (https://www.r-project.org, accessed on 2 February 2023) based on the risk factors concluded from the multivariable analysis, including SOS and clinical characteristics. The concordance index (C-index) was measured by comparing the predicted survival with the observed survival probability to better clarify the independent discrimination performance of constructed nomogram. The larger the C-index, the more accurate the prognostic stratification [22]. The calibration was assessed by a calibration curve. The standard curve is a straight line passing through the origin of the coordinate axis with a slope of 1. If the predicted calibration curve is closer to the standard curve, the better predictive ability of the nomogram will be. The cut-off values for total risk points were assessed by the X-tile software, and patients were divided into high-risk and low-risk groups based on the cut-off value of risk scores. Patients’ RFS were analyzed by using the Kaplan-Meier method, and the differences were compared by log-rank test. Statistical analyses were conducted by using SPSS 23.0 (IMB-SPSS Inc., Armonk, NY, USA), and all survival curves were constructed by R version 4.1.1 software (https://www.r-project.org, accessed on 2 February 2023).

## 3. Results

### 3.1. Clinical Characteristics of Patients

A total of 955 patients with non-small cell lung cancer were included in this study. Among them, 407 (42.6%) were males, 548 (57.4%) were females, and 82.9% of the patients had no smoking history. Of the patients, 90.4% underwent lobectomy, and the main pathology type of patients was Acinar/Papillary (58.8%). EGFR mutations were not exhibited in 358 (37.5%) patients, and 597 (62.5%) patients showed EGFR mutations. The main predominant EGFR mutation types were 19-del (27.9%) and L858R (29.2%). The levels of oxidative stress-related biochemical markers (TBIL, DBIL, ALB, UA, CRE, LDH) and other clinical characteristics are shown in Table 1. The optimal cut-off values of biochemical indicators and the corresponding K-M survival analysis were determined by X-tile software (Figure A1).

### 3.2. Construction of the Systematic Oxidative Stress Score

To explore the prognostic value of the systemic oxidative stress score, we determined the optimal cut-off values of the relevant systemic oxidative stress indicators according to the X-tile software and performed the dichotomous conversion. The cut-off values of systemic oxidative stress indicators were as follows: TBIL: 7.9 μmol/L, DBIL: 2.9 μmol/L, ALB: 38 g/L, UA: 325 μmol/L, CRE: 58 μmol/L, and LDH: 198 U/L (Figure A1). To extend the analysis, we included candidate indicators with *p*-values less than 0.2 in the univariable Cox regression analysis into the multivariable Cox regression analysis, and the results showed that UA (HR: 0.566, 95% CI: 0.373–0.858, *p* = 0.007), CRE (HR: 1.587, 95% CI: 1.051–2.395, *p* = 0.028) and LDH (HR: 1.991, 95% CI: 1.233–3.214, *p* = 0.005) were independent prognostic indicators associated with RFS of patients with NSCLC (Table 2). To construct the SOS system, we re-performed the multivariable Cox regression analysis on the screened candidates and calculated the corresponding regression coefficient. Finally, the SOS formula was as follows: SOS = 0.490*CRE + 0.636*LDH − 0.562*UA (Table 3).

### 3.3. Survival Analysis and the Relationship between SOS and Clinical Characteristics

Based on the constructed SOS formula, we calculated the SOS values for all patients and determined the optimal cut-off value of 15.1 using the X-tile (Figure A2a). We classified patients with SOS greater than 15.1 as the high-level SOS group and vice versa as the low-level SOS group. K-M analysis showed that patients with high-level SOS had significantly worse RFS than those with low-level SOS (Figure 1, *p* = 0.0022). The relationship between SOS and clinical characteristics is shown in Table 4. The results showed that female patients, the non-smoking group, and patients with tumor size greater than 1 cm tended to have higher SOS, and EGFR mutation had no association with SOS (*p* = 0.806).

### 3.4. SOS Was an Independent Prognostic Indicator of RFS for NSCLC Patients

After obtaining the SOS, we combined the clinical characteristics and SOS of the patients and performed univariable and multivariable Cox regression analyses of the relevant factors. The results showed that SOS, VPI, predominant pattern, and age at surgery were independent prognostic indicators of RFS in NSCLC patients. Patients with high-level SOS had worse survival (HR: 2.015, 95% CI: 1.229–3.303, *p* = 0.005) (Table 5). Notably, EGFR mutations were not an independent prognostic indicator of RFS in patients.

### 3.5. Construction and Validation of the Prognostic Nomogram for NSCLC Patients

Based on the results of the multivariable Cox regression analysis, we constructed a prognostic nomogram of the 3-year and 5-year RFS predictions model for NSCLC patients (Figure 2), and the internal calibration curves performed well (Figure 3a,b), with the C-index value of 0.684, which indicated that the accuracy of model performed well. Finally, we calculated the scores of all patients based on the prognostic nomogram and used X-tile to evaluate the optimal cut-off value. Patients were divided into high-risk and low-risk groups according to the optimal cut-off value (Figure A2b). The K-M survival analysis showed that patients in the low-risk group had better survival (Figure 4, *p* < 0.0001).

## 4. Discussion

At present, lung cancer gradually tends to be tumors with early stage and small invasive size with the promotion of chest CT [3,4], but the prognosis of these patients is still highly heterogeneous, and there is still the possibility of recurrence after radical resection of the primary tumor [7]. Therefore, it is necessary to accurately assess the risk of postoperative recurrence in these patients and identify the high-risk group. Previous studies have proposed that ROS is associated with the development of a variety of tumors [15,22], and SOS based on the biomarkers of systematic oxidative stress in peripheral blood, including CRE, TBIL, LDH, BUN, and ALB, have yielded adverse results in the prognostication of breast cancer [23], but their role in the prognosis of patients with early-stage NSCLC is unclear.

In the present study, we analyzed the patients’ data from the database of Shanghai Pulmonary Hospital to investigate whether these biomarkers have predictive value in the patient’s prognosis. For better comparison, the continuous variables in this study were converted to categorical variables by cut-off values. According to the results of multivariable Cox regression analysis, three independent prognostic factors for RFS, including UA, CRE, and LDH, were identified. We found that elevated UA was associated with better RFS in patients with stage I LUAD, while LDH and CRE were associated with a poorer prognosis, which was consistent with previous studies [20,21,24,25,26,27]. In order to combine these factors, SOS was established based on these three indicators. Multivariable Cox regression analysis among the clinical characteristics and SOS showed that four independent prognostic factors for RFS, including age at surgery, VPI, predominant pattern, and SOS, were closely related to the prognosis of patients, and higher SOS was related to poorer survival. Then, a nomogram based on SOS and clinical characteristics was built, which could provide high accuracy in predicting RFS. In the model, we observed that the predominant pattern of the tumor, especially micropapillary/solid subtypes had the greatest impact on patients’ outcomes, which was consistent with the previous studies [28,29]. The C-index and calibration curves were used to verify the accuracy and predictive ability of the model. We found that the C-index of the nomogram based on SOS was 0.684, and the calibration curves of the model for 3-year and 5-year RFS showed an optimal agreement between prediction and actual observation. Furthermore, patients were categorized into high-risk and low-risk groups with significant differences in RFS based on the cut-off value of risk scores, which also verified the clinical practicability of the model.

TBIL, DBIL, and ALB are all indicators that reflect the function of the liver. Previous studies have shown that TBIL and DBIL are associated with the prognosis of tumors, but most of those studies are focused on gastrointestinal tumors [30,31,32]. For patients with advanced NSCLC, the prognostic role of DBIL has also been proved [33], but no relationship between TBIL and DBIL and the prognosis of patients with stage I LUAD was observed in our study. In addition to reflecting the nutritional status of the human body, ALB has been proposed to be an endogenous antioxidant that can reduce cancer risk by exerting anti-cancer properties [34,35]. However, there were also several studies reporting nonsignificant associations between ALB and the prognosis of lung cancer [36,37], which is in accord with our findings. We consider the primary reason is that patients with advanced lung cancer and gastrointestinal tumors often have poor nutritional status due to cachexia, resulting in significant changes in liver-related blood markers. However, this phenomenon is rarely seen in patients with early-stage NSCLC. STAS has also previously been shown to be a risk factor for the postoperative prognosis of patients with stage I NSCLC [38,39,40,41], but no significant difference in survival was observed in this study. This may be due to the fact that STAS could be easily confused with artifacts such as loose tumor tissue fragments, which led to a bias toward the diagnosis [42,43]. Moreover, we found that patients who underwent sub-lobectomy had a comparable prognosis to those who underwent lobectomy in our study, which is consistent with the results of recent studies [44,45,46,47,48].

There are many studies on the mechanism of oxidative stress involved in the formation and development of tumors, but relatively few reports on its effect on the prognosis of patients with tumors. Notably, our study found the prognostic role of SOS in patients with early-stage NSCLC, and the nomogram based on the SOS could accurately stratify the risk of tumor recurrence after surgery in patients with stage I LUAD. This study can help clinicians identify patients with poor prognoses and provide more treatment options and postoperative follow-up for high-risk patients. In addition, the results of this study are conducive to the subsequent study of the relationship between oxidative stress and tumor prognosis and provide a reference for the development of targeted therapy for oxidative stress.

It should be noted that there were still some limitations in our study. First of all, the data of patients were all from a single-center database without an external validation cohort, which resulted in a lack of representativeness. Furthermore, as a retrospective study, inevitable bias was also caused to some extent. Secondly, for early-stage NSCLC, the follow-up duration of this study was relatively short, leading to the number of deaths observed after surgery was rare, and our study was only able to investigate factors associated with RFS of patients after surgery. Finally, due to the lack of effective methods for the detection of ROS, our study could only indirectly reflect the status of oxidative stress through some indicators in the blood, but the test items were decided by each clinician, and there was a certain subjectivity, resulting in the indicators may not be comprehensive enough. Additionally, our study has preliminarily found that ROS is associated with prognosis in patients with early-stage lung adenocarcinoma, but the specific mechanism of ROS’s role in tumor proliferation and apoptosis needs to be further explored in future research. Therefore, multicenter and prospective observational studies are needed to collect more biomarkers and samples to validate the accuracy and practicability of SOS in predicting the prognosis of patients with stage I LUAD after surgery.

## 5. Conclusions

In summary, SOS was significantly associated with the poor prognosis of patients with stage I LUAD based on the status of oxidative stress. In addition, the established nomogram combined with SOS could predict the prognosis of patients accurately.

## Figures and Tables

**Figure 1 cancers-15-01718-f001:**
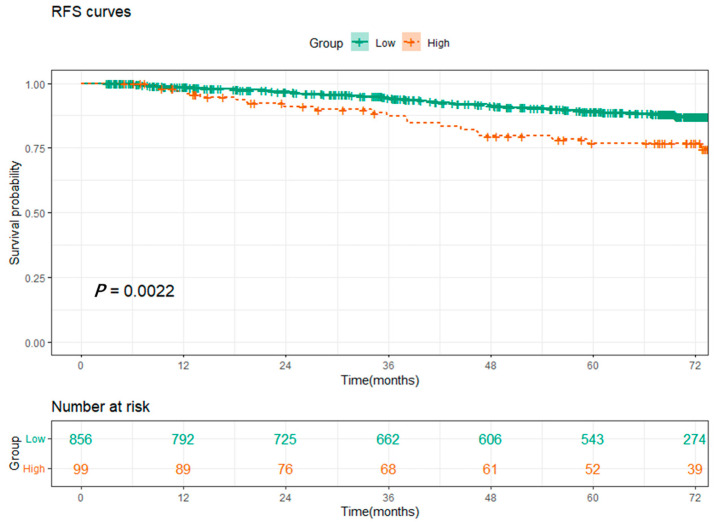
Kaplan–Meier survival curves for patients between the low-level (≤15.1) and high-level (>15.1) groups of systematic oxidative stress scores.

**Figure 2 cancers-15-01718-f002:**
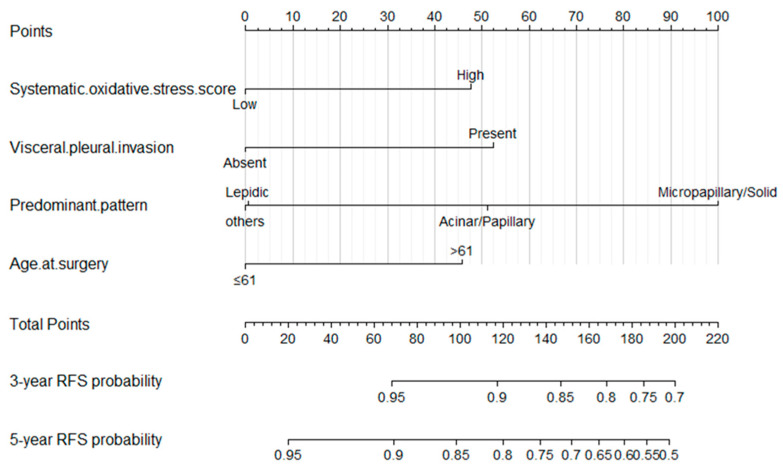
Nomogram for 3- and 5-year recurrence-free survival in patients with stage I lung adenocarcinoma.

**Figure 3 cancers-15-01718-f003:**
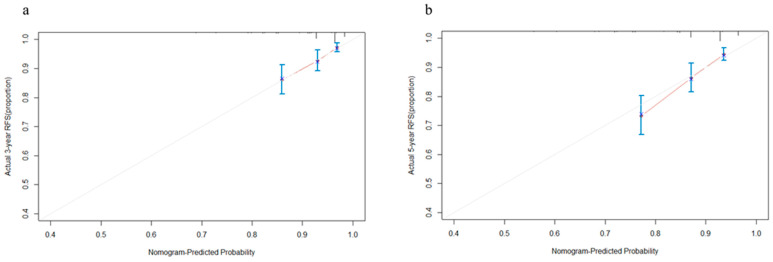
Calibration curves of the nomogram for the 3- (**a**) and 5-year (**b**) recurrence-free survival prediction of patients with stage I lung adenocarcinoma. The abscissa is the predicted probability, and the ordinate is the actual observed rate. The dashed diagonal line is the reference line, which is the case when the predicted value is equal to the actual value. The line between the blue dots is the 95% confidence interval, and the red line is the curve of the actual probability of occurrence. Different colors can show the results of the study more clearly.

**Figure 4 cancers-15-01718-f004:**
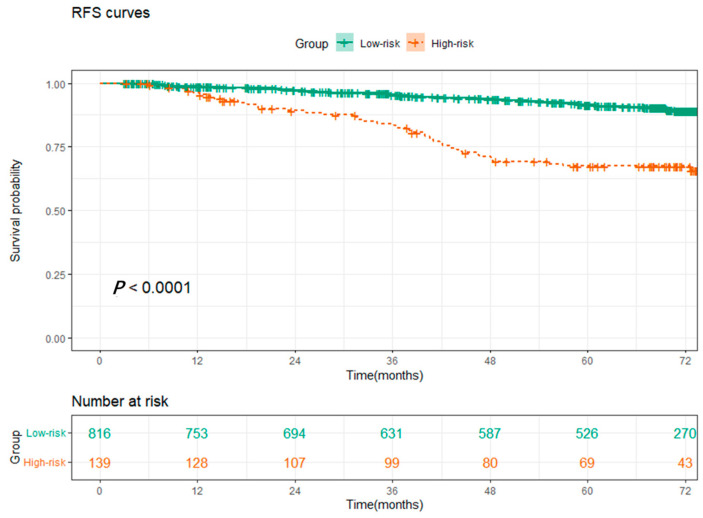
Kaplan–Meier survival curves for recurrence-free survival between the high-risk and low-risk groups based on the nomogram.

**Table 1 cancers-15-01718-t001:** Baseline clinicopathologic characteristics of patients.

Variables	*n* = 955
Gender	
Male	407 (42.6%)
Female	548 (57.4%)
Age at surgery, years (IQR)	61 (56–66)
≤61	491 (51.4%)
>61	464 (48.6%)
Smoking history	
No	792 (82.9%)
Yes	163 (17.1%)
Extent of surgery	
Lobectomy	863 (90.4%)
Sub-lobectomy	91 (9.5%)
Pneumonectomy	1 (0.1%)
Predominant pattern	
Lepidic	302 (31.6%)
Acinar/Papillary	562 (58.8%)
Micropapillary/Solid	59 (6.2%)
Others	32 (3.4%)
Tumor size, cm	
≤1.0	72 (7.5%)
1.1–2.0	468 (49.0%)
2.1–3.0	316 (33.1%)
3.1–4.0	99 (10.4%)
Visceral pleural invasion	
Absent	861 (90.2%)
Present	94 (9.8%)
Lymphovascular invasion	
Absent	945 (99.0%)
Present	10 (1.0%)
Spread through air space	
Absent	925 (96.9%)
Present	30 (3.1%)
Epidermal growth factor receptor mutation	
Without	358 (37.5%)
19-del	266 (27.9%)
L858R	279 (29.2%)
Others	52 (5.4%)
Adjuvant chemotherapy	
No	683 (71.5%)
Yes	272 (28.5%)
Pathological stage	
IA	775 (81.2%)
IB	180 (18.8%)
Total bilirubin	
≤7.9	128 (13.4%)
>7.9	827 (86.6%)
Direct bilirubin	
≤2.9	150 (15.7%)
>2.9	805 (84.3%)
Albumin	
≤38	175 (18.3%)
>38	780 (81.7%)
Uric acid	
≤325	560 (58.6%)
>325	395 (41.4%)
Creatinine	
≤58	395 (41.4%)
>58	560 (58.6%)
Lactate dehydrogenase	
≤198	838 (87.7%)
>198	117 (12.3%)

Data are expressed as *n* (%) or median (IQR). IQR, interquartile range.

**Table 2 cancers-15-01718-t002:** Univariable and multivariable analyses for recurrence-free survival based on the biochemical indicators.

Variables	Univariable Analysis			Multivariable Analysis		
	HR	95% CI	*p*-Value	HR	95% CI	*p*-Value
Total bilirubin						
≤7.9	1					
>7.9	1.636	0.854–3.136	0.138			
Direct bilirubin						
≤2.9	1			1		
>2.9	1.586	0.871–2.890	0.132	1.681	0.918–3.077	0.093
Albumin						
≤38	1					
>38	1.384	0.814–2.352	0.230			
Uric acid						
≤325	1			1		
>325	0.674	0.452–1.004	0.053	0.566	0.373–0.858	0.007
Creatinine						
≤58	1			1		
>58	1.378	0.929–2.043	0.111	1.587	1.051–2.395	0.028
Lactate dehydrogenase						
≤198	1			1		
>198	1.855	1.152–2.986	0.011	1.991	1.233–3.214	0.005

**Table 3 cancers-15-01718-t003:** Multivariable Cox regression analysis of candidate indicators for systematic oxidative stress scores.

	Multivariable Analysis			
Variables	Coef	Exponential (Coef)	95% CI	*p*-Value
Uric acid	−0.562	0.570	0.376–0.865	0.008
Creatinine	0.490	1.632	1.081–2.461	0.020
Lactate dehydrogenase	0.636	1.888	1.173–3.041	0.009

**Table 4 cancers-15-01718-t004:** Characteristics of patients in low-level and high-level groups of systematic oxidative stress scores.

Variables	Total (*n* = 955)	High-Level (*n* = 99)	Low-Level (*n* = 856)	*p*-Value
Gender				<0.001
Male	407	15 (15.2%)	392 (45.8%)	
Female	548	84 (84.8%)	464 (54.2%)	
Age at surgery, years				0.408
≤61	491	47 (47.5%)	444 (51.9%)	
>61	464	52 (52.5%)	412 (48.1%)	
Smoking history				0.005
No	792	92 (92.9%)	700 (81.8%)	
Yes	163	7 (7.1%)	156 (18.2%)	
Predominant pattern				0.942
Lepidic	302	34 (34.3%)	268 (31.3%)	
Acinar/Papillary	562	56 (56.6%)	506 (59.1%)	
Micropapillary/Solid	59	6 (6.1%)	53 (6.2%)	
Others	32	3 (3.0%)	29 (3.4%)	
Tumor size, cm				0.012
≤1.0	72	3 (3.0%)	69 (8.1%)	
1.1–2.0	468	39 (39.4%)	429 (50.1%)	
2.1–3.0	316	41 (41.4%)	275 (32.1%)	
3.1–4.0	99	16 (16.2%)	83 (9.7%)	
Visceral pleural invasion				0.246
Absent	861	86 (86.9%)	775 (90.5%)	
Present	94	13 (13.1%)	81 (9.5%)	
Lymphovascular invasion				0.127
Absent	945	96 (97.0%)	849 (99.2%)	
Present	10	3 (3.0%)	7 (0.8%)	
Spread through air space				1
Absent	925	96 (97.0%)	829 (96.8%)	
Present	30	3 (3.0%)	27 (3.2%)	
Epidermal growth factor receptor mutation				0.806
Without	358	36 (36.4%)	322 (37.6%)	
19-del	266	25 (25.2%)	241 (28.2%)	
L858R	279	33 (33.3%)	246 (28.7%)	
Others	52	5 (5.1%)	47 (5.5%)	
Extent of surgery				0.841
Lobectomy	863	89 (89.9%)	774 (90.4%)	
Sub-lobectomy	91	10 (10.1%)	81 (9.5%)	
Adjuvant chemotherapy				0.110
No	683	64 (64.6%)	619 (72.3%)	
Yes	272	35 (35.4%)	237 (27.7%)	

**Table 5 cancers-15-01718-t005:** Univariable and multivariable analyses for recurrence-free survival based on the systematic oxidative stress score and clinical characteristics.

	Univariable Analysis			Multivariable Analysis		
Variables	HR	95% CI	*p*-Value	HR	95% CI	*p*-Value
Gender						
Male	1					
Female	0.796	0.546–1.159	0.233			
Age at surgery, years (IQR)						
≤61	1			1		
>61	1.931	1.308–2.850	0.001	2.003	1.350–2.971	0.001
Smoking history						
No	1					
Yes	0.992	0.599–1.645	0.976			
Extent of surgery						
Lobectomy	1					
Sub-lobectomy	1.021	0.533–1.958	0.950			
Predominant pattern						
Lepidic	1			1		
Acinar/Papillary	2.206	1.318–3.692	0.003	2.103	1.247–3.548	0.005
Micropapillary/Solid	4.713	2.375–9.352	<0.001	4.022	1.961–8.250	<0.001
Others	0.874	0.203–3.769	0.857	0.997	0.231–4.305	0.996
Tumor size, cm						
≤1.0	1					
1.1–2.0	2.126	0.659–6.859	0.207			
2.1–3.0	3.880	1.209–12.449	0.023			
3.1–4.0	3.650	1.057–12.611	0.041			
Visceral pleural invasion						
Absent	1			1		
Present	2.839	1.816–4.440	<0.001	2.198	1.384–3.489	0.001
Lymphovascular invasion						
Absent	1					
Present	5.858	2.384–14.393	<0.001			
Spread through air space						
Absent	1					
Present	2.045	0.898–4.660	0.089			
Epidermal growth factor receptor mutation						
Without	1					
19-del	0.966	0.610–1.529	0.882			
L858R	0.835	0.520–1.341	0.455			
Others	0.972	0.414–2.280	0.947			
Adjuvant chemotherapy						
No	1					
Yes	1.274	0.857–1.895	0.231			
Systematic oxidative stress score						
Low	1			1		
High	2.097	1.291–3.407	0.003	2.015	1.229–3.303	0.005

## Data Availability

The data sets are available from the corresponding authors upon reasonable request.

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
