# Peer review of "A Novel Systematic Oxidative Stress Score Predicts the Survival of Patients with Early-Stage Lung Adenocarcinoma"

_cancers, 2023, doi:10.3390/cancers15061718_

Round 1
Reviewer 1 Report
Qian et al reported their work named "A Novel Systematic Oxidative Stress Score Predicts the Survival of Patients with Early-stage Lung Adenocarcinoma" and concluded that "SOS is an independent prognostic indicator for stage I LUAD. Besides, the constructed nomogram based on SOS could accurately predict the survival of those patients.". I have some comments:
- In your sentence "continuous variables were tested by using a t-test.", please test continuous variables for normality then report median and IQR and test between group difference using Mann Whitney test if they were not normally distributed while report mean and SD and test between group difference using t test.
- Please expand on how you select cutoffs that you mentioned in this sentence "The cut-off values for RFS of six oxidative stress-related biochemical indicators including TBIL, DBIL, ALB, UA, CRE, and LDH were calculated by X-tile software".
- Please add reference to this sentence "The concordance index (C-index) was measured by comparing the predicted survival with the observed survival probability to better clarify the independent discrimination performance of constructed nomogram".
- Please keep only one statistical program, the one used for nomogram and KM curves. Please try to add the used R code to supplements. It is advisable to create "dynamic nomogram" that could be one using shiny package in R as in the following https://pubmed.ncbi.nlm.nih.gov/34468733/.
- Please avoid abbreviations in the tables and figures legend.
- Table 3: Please revisit the title and the outcome you are predicitng using this multivariate analysis. Are you predicting SOS? You are reporting hazard ratio which is for time to events outcome. Please delete "Coef" column.
Table 4: Please add criteria of the entire cohort to the left of your "high-level SOS (n=99)" column.
- Table 5: Please specify why you did not include tumor size and LVI into your multivariate model although they were significant in univariate analysis. What are your variables selection criteria for the multivariate analysis and what is the goodness of fit for your model?
- Please specify the validation method in your statistical analysis section.
Author Response
1. In your sentence "continuous variables were tested by using a t-test.", please test continuous variables for normality then report median and IQR and test between group difference using Mann Whitney test if they were not normally distributed while report mean and SD and test between group difference using t test.
Reply: Thank you very much for your comments, we have revised the methods we compare variables.
2. Please expand on how you select cutoffs that you mentioned in this sentence "The cut-off values for RFS of six oxidative stress-related biochemical indicators including TBIL, DBIL, ALB, UA, CRE, and LDH were calculated by X-tile software".
Reply: The outcome in the X-tile software was defined as tumor recurrence, and the 6 oxidative stress-related indicators in the study were used as markers respectively, and then analyzed. By clicking on 2Pop X-tile Plot, the optimal cut-off value can be automatically found, and the data can be divided into high-level and low-level groups.
3. Please add reference to this sentence "The concordance index (C-index) was measured by comparing the predicted survival with the observed survival probability to better clarify the independent discrimination performance of constructed nomogram".
Reply: Thank you very much for your comments, we have added the reference to the sentence in the manuscript.
4. Please keep only one statistical program, the one used for nomogram and KM curves. Please try to add the used R code to supplements. It is advisable to create "dynamic nomogram" that could be one using shiny package in R as in the following https://pubmed.ncbi.nlm.nih.gov/34468733/.
Reply: Thank you very much for your comments. In this study, we did only use R software alone for plotting nomogram and K-M curves. SPSS was only used for early data analysis. If needed, the relevant code can be provided by email.
5. Please avoid abbreviations in the tables and figures legend.
Reply: Thank you very much for your comments, we've made adjustments to all legends.
6. Table 3: Please revisit the title and the outcome you are predicting using this multivariate analysis. Are you predicting SOS? You are reporting hazard ratio which is for time to events outcome. Please delete "Coef" column.
Reply: Thank you very much for your comments. In our study, the systematic oxidative stress score was calculated based on the lowest Akaike information criterion (AIC) value, and “Coef” gave these three biomarkers different values.
7. Please add criteria of the entire cohort to the left of your "high-level SOS (n=99)" column.
Reply: Thank you very much for your comments, we've made adjustments to the table.
8. Table 5: Please specify why you did not include tumor size and LVI into your multivariate model although they were significant in univariate analysis. What are your variables selection criteria for the multivariate analysis and what is the goodness of fit for your model?
Reply: Thank you very much for your comments. In fact, we included LVI and tumor size in the multivariable analysis, but because of the sample size, these two factors did not show significant significance in the final results of multivariable regression. Besides, because we used the backward method of the stepwise regression method, these two factors are not shown in the table at the end.
To extend the analysis, variables with P < 0.2 in the univariable Cox regression analysis were eventually included in the multivariable Cox regression analysis to obtain independent prognostic predictors associated with the RFS of patients. Statistical significance was considered as a P < 0.05 on two sides.
Omnibus tests suggest that the model with SOS has advantages over the model without SOS.
9. Please specify the validation method in your statistical analysis section.
Reply: Thank you very much for your comments, we have listed the methods of validation in the manuscript. Calibration curves for RFS probabilities showed a good agreement between the nomogram prediction and actual observation. Furthermore, the C-index of the nomogram showed the accuracy of prognostic stratification Additionally, the patients were divided into two groups according to cut-off values of risk points based on the nomogram, and a significant difference in RFS was served between the high-risk and low-risk groups (P<0.001), which also validated the accuracy of the model.
Reviewer 2 Report
This is an interesting retropsective study on Oxidative Stress Score that could Predict the survival of patients with early-stage lung adenocarcinoma. The study design is very well defined and the results very interesting. In the methods authors should define if patients have been evaluated for receiving any special diet or antioxidant therapy as support taht could bias the results. Overall the quality is high and the definition of a nomagram is very important and could help in future to open the way to propsective trials.
Author Response
Thank you very much for your comments, we feel that this suggestion is very pertinent and can also make our research more rigorous. We have included this standard in the revised version.
Reviewer 3 Report
I would like to provide a certain comment:Biological markers were chosen by the authors rather randomly. They don't give any interest to very promising markers of NSCLC development and surviving prediction as peroxiredoxins, particularly PRDX2 invoved in the regulation of cellular ROS in normal and cancer cells (Chen L et al. Onco Targets Ther. 2018;11:8381-8388; Chen Y, et al..Biomed Res Int.;2020:8359860; doi: 10.2147/OTT.S181314; Jing X. et al. Biomed Res Int. 2020. 3;2020; doi: 10.1155/2020/1276328.; Mei, J. et al. Biomark Res. 2019. 7, 16. doi.org/10.1186/s40364-019-0168-9SIDE)
Author Response
Thank you very much for your comments. We would like to make it clear that the study did not select ROS-related indicators completely randomly. Based on previous studies (1-3), combined with the data from Shanghai Pulmonary Hospital, we included as many ROS-related indicators as possible.
As for the PRDX2 you mentioned, it has indeed been shown to play an important role in tumor proliferation and apoptosis as a member of the peroxiredoxin family of antioxidant enzymes, and its mechanism of action has also been well explored. However, our study aimed to predict the prognosis of patients with early-stage lung adenocarcinoma through clinically readily available ROS-related indicators, but PRDX2 does not seem to be so readily available compared with these indicators. Also, there may also be some link between these indicators and PRDX2, which has not yet been explored.
Our study has preliminarily found that ROS is associated with prognosis in patients with early-stage lung adenocarcinoma, but the specific mechanism of ROS's role in tumor proliferation and apoptosis needs to be further explored in future research.
- Zhang K, Ping L, Du T, Wang Y, Sun Y, Liang G, et al. A Novel Systematic Oxidative Stress Score Predicts the Prognosis of Patients with Operable Breast Cancer. Oxid Med Cell Longev. 2021;2021:9441896.
- Sandesc M, Rogobete AF, Bedreag OH, Dinu A, Papurica M, Cradigati CA, et al. Analysis of oxidative stress-related markers in critically ill polytrauma patients: An observational prospective single-center study. Bosn J Basic Med Sci. 2018;18(2):191-7.
- Periasamy S, Hsu DZ, Fu YH, Liu MY. Sleep deprivation-induced multi-organ injury: role of oxidative stress and inflammation. EXCLI J. 2015;14:672-83.
Round 2
Reviewer 1 Report
Thanks for your edits and it is my pleasure to accept your work.
Minor post-acceptance edits:
Please change HR in table 3 to Exponential(coef) as HR is used only for time to events outcome as survival which is not the case in this table.
Reviewer 3 Report
Explanation of the authors considering the choise of the clinically readily available ROS-related indicators, as comparable to not available PRDX2, as the prediction for patients with early-stage lung adenocarcinoma prognosis, is acceptable. Manuscript can be published in the present form. However mentioning of PRDX2 as valuable ROS -related indicators could be interested for readers as subject for further studies.